# Shyft v4.8: A Framework for Uncertainty Assessment and Distributed Hydrologic Modelling for Operational Hydrology

John F. Burkhart[1,2], Felix N. Matt[1,2], Sigbjørn Helset[2], Yisak Sultan Abdella[2], Ola Skavhaug[3], and Olga Silantyeva[1]

[1]Department of Geosciences, University of Oslo, Oslo, Norway
[2]Statkraft AS, Lysaker, Norway
[3]Expert Analytics AS, Oslo, Norway

**Correspondence:** John F. Burkhart (john.burkhart@geo.uio.no)

**Abstract.** This paper presents Shyft, a novel hydrologic modelling software for streamflow forecasting targeted for use in hydropower production environments and research. The software enables the rapid development and implementation in operational settings, the capability to perform distributed hydrologic modelling with multiple model and forcing configurations. Multiple models may be built up through the creation of hydrologic algorithms from a library of well known routines or through the creation of new routines, each defined for processes such as: evapotranspiration, snow accumulation and melt, and soil water response. Key to the design of Shyft is an Application Programming Interface (api) that provides access to all components of the framework (including the individual hydrologic routines) via Python, while maintaining high computational performance as the algorithms are implemented in modern C++. The api allows for rapid exploration of different model configurations and selection of an optimal forecast model. Several different methods may be aggregated and composed, allowing direct inter-comparison of models and algorithms. In order to provide an enterprise level software, strong focus is given to computational efficiency, code quality, documentation and test coverage. Shyft is released Open Source under the GNU Lesser General Public License v3.0 and available at https://gitlab.com/shyft-os, facilitating effective cooperation between core developers, industry, and research institutions.

## 1 Introduction

Operational hydrologic modelling is fundamental to several critical domains within our society. For the purposes of flood prediction and water resource planning, the societal benefits are clear. Many nations have hydrological services that provide water-related data and information in a routine manner. The World Meteorological Organization gives an overview of the responsibilities of these services and the products they provide to society including: monitoring of hydrologic processes, provision of data, water-related information including seasonal trends and forecasts, and importantly, decision-support services (World Meteorological Organization, 2006).

Despite the abundantly clear importance of such operational systems, implementation of robust systems that are able to fully incorporate recent advances of remote sensing, distributed data acquisition technologies, high resolution weather model inputs, or ensembles of forecasts remains a challenge. Pagano et al. (2014) provide an extensive review of these challenges,

as well as the potential benefits afforded by overcoming some relatively simple barriers. The Hydrologic Ensemble Prediction EXperiment (https://www.hepex.org) is an activity that has been ongoing since 2004, and there is extensive research on the importance of the role of ensemble forecasting to reduce uncertainty in operational environments (e.g. Pappenberger et al. (2016); Wu et al. (2020))

As most operational hydrological services are within the public service, government policies and guidelines influence the area of focus. Recent trends show efforts towards increasing commitment to sustainable water resource management, disaster avoidance and mitigation, and the need for Integrated Water Resource Management as climatic and societal changes are stressing resources.

For hydropower production planning, operational hydrologic modelling provides the foundation for energy market forecasting and reservoir management, addressing both the interest of the power plant operator as well as governmental regulations. Hydropower production accounts for 16% of the world's electricity generation and is the leading renewable source for electricity (non-hydro renewable and waste sum up to about 7%). Between 2007 and 2015, the global hydropower capacity increased by more than 30% (World Energy Council, 2016). In many regions around the globe, hydropower is therefore playing a dominant role in the regional energy supply. In addition, as energy production from renewable sources with limited managing possibilities (e.g. from wind and solar) grows rapidly, hydropower production sites equipped with pump storage systems provide the possibility to store energy efficiently at times when total energy production surpasses demands. Increasingly critical to the growth of energy demand is the proper accounting of water use and information to enable water resource planning (Grubert and Sanders, 2018).

Great advances in hydrologic modelling are being made in several facets: new observations are becoming available through novel sensors (McCabe et al., 2017), Numerical Weather Prediction (NWP) and reanalysis data are increasingly reliable (Berg et al., 2018), detailed estimates of Quantitative Precipitation Estimates (QPEs) are available as model inputs (Moreno et al., 2012, 2014; Vivoni et al., 2007; Germann et al., 2009; Liechti et al., 2013), improved algorithms and parameterizations of physical processes (Kirchner, 2006), and perhaps most significantly, we have advanced greatly in our understanding of uncertainty and the quantification of uncertainty within hydrologic models (Westerberg and McMillan, 2015; Teweldebrhan et al., 2018c). Anghileri et al. (2016) evaluated the forecast value of long-term inflow forecasts for reservoir operations using the approach of Ensemble Streamflow Prediction (ESP) (Day, 1985). Their results show that the value of a forecast using ESP varies significantly as a function of the seasonality, hydrologic conditions, and reservoir operation protocols. Regardless, having a robust ESP system in place allows operational decisions that will create value. In a follow-on study, Anghileri et al. (2019) showed that pre-processing of meteorological input variables can also significantly benefit the forecast process.

A significant challenge remains, however, in environments that have operational requirements. In such an environment, 24/7 up-time operations, security issues, and requirements from Information Technology departments often challenge introducing new or 'innovative' approaches to modelling. Furthermore, there is generally a requirement to maintain an existing model configuration while exploring new possibilities. Often, the implementation of two parallel systems is daunting and presents a technical roadblock. An example of the scale of the challenge is well defined in Zappa et al. (2008) where the author's contributions to the results of the Demonstration of Probabilistic Hydrological and Atmospheric Simulation of flood Events in the

Alpine region (D-PHASE) project under the Mesoscale Alpine Programme (MAP) of the WMO World Weather Research Program (WWRP) are highlighted. In particular, they had the goal to operationally implement and demonstrate a new generation of flood warning systems in which each catchment had one or more hydrological models implemented. However, following the 'demonstration' period, "no MAP D-PHASE contributor was obviously able to implement its hydrological model in all basins and couple it with all available deterministic and ensemble numerical weather prediction (NWP) models."; presumably resulting from the complexity of the configurations required to run multiple models with differing domain configurations, input file formats, operating system requirements, and so forth.

There is an awareness in the hydrologic community regarding the nearly profligate abundance of hydrologic models. Recent efforts have proposed the development of a community based hydrologic model (Weiler and Beven, 2015). The WRF-Hydro platform (Gochis et al., 2018) is a first possible step in that direction, along with SUMMA (Clark et al., 2015a) a highly configurable and flexible platform for the exploration of structural model uncertainty. However, the WRF-Hydro platform is computationally excessive for many operational requirements and SUMMA was designed with different objectives in mind that what has been developed within Shyft. For various reasons (see Section 1.2) the development of Shyft was initiated to fill a gap in operational hydrologic modeling.

Shyft is a modern cross-platform Open Source toolbox that provides a computation framework for spatially distributed hydrologic models suitable for inflow forecasting for hydropower production. The software is developed by Statkraft AS, Norway's largest hydropower company and Europe's largest generator of renewable energy in cooperation with the research community. The overall goal for the toolbox is to provide Python-enabled high performance components with industrial quality and use in operational environments. Purpose built for production planning in a hydropower environment, Shyft provides tools and libraries that also aim for domains other than hydrolgic modelling, including modelling energy markets and high performance time-series calculations, which won't be discussed herein.

In order to target hydrologic modelling, the software allows the creation of *model stacks* from a library of well known hydrologic routines. Each of the individual routines are developed within Shyft as a module, and are defined for processes such as: evapotranspiration, snow accumulation and melt, or soil water response. Shyft is highly extensible, allowing others to contribute or develop their own routines. Other modules can be included in the model stack for improved handling of snow melt or to pre-process and interpolate point input timeseries of temperature and precipitation (for example) to the geographic region. Several different methods may be easily aggregated and composed, allowing direct intercomparison of algorithms. The method stacks operate on a one-dimensional geolocated 'cell', or a collection of cells may be constructed to create catchments and regions within a domain of interest. Calibration of the methods can be conducted at the cell, catchment, or region level.

The objectives of Shyft are to: (i) provide a flexible hydrologic forecasting toolbox built for operational environments, (ii) enable computationally efficient calculations of hydrologic response at the regional scale, (iii) allow for using the multiple working hypothesis to quantify forecast uncertainties, (iv) provide the ability to conduct hydrologic simulations with multiple forcing configurations, and (v) foster rapid implementation into operational modeling improvements identified through research activities.

To address the first and second objectives, computational efficiency and well test-covered software have been paramount. Shyft is inspired by research software developed for testing multiple working hypothesis (Clark et al., 2011). However, the developers felt, that more modern coding standards and paradigms could provide significant improvements in computational efficiency and flexibility. Using the latest C++ standards, a templated code concept was chosen in order to provide flexible software for use in business critical applications. As Shyft is based on advanced templated C++ concepts, the code is highly efficient and able to take advantage of modern day compiler functionality, minimizing risk of faulty code and memory leaks. To address the latter two objectives, the templated language functionality allows for the development of different algorithms that are then easily implemented into the framework. An Application Programming Interface (api) is provided for accessing and assembling different components of the framework, including the individual hydrologic routines. The api is exposed to both the C++ and Python languages allowing for rapid exploration of different model configurations and selection of an optimal forecast model. Multiple use cases are enabled through the api. For instance, one may choose to explore parameter sensitivity of an individual routine directly, or one may be interested purely in optimized hydrologic prediction, in which case one of the predefined and optimized *model stacks*, a sequence of routines forming a hydrologic model, would be of interest.

The goal of this paper is two-fold: to introduce Shyft, and to demonstrate some recent applications that have used heterogeneous data to configure and evaluate the fidelity of simulation. First, we present the core philosophical design decisions in Section 2 and provide and overview of the architecture in Section 3. The model formulation and hydrologic routines are discussed in Sections 4 and 5. Secondly, we provide a review of several recent applications that have addressed issues of uncertainty, evaluated satellite data forcing, and explore data assimilation routines for snow.

### 1.1 Other frameworks

To date, a large number of hydrological models exist, each differing in the input data requirements, level of details in process representation, flexibility in the computational subunit structure, and availability of code and licensing. In the following we provide a brief summary of several models that have garnered attention and a user community, but ultimately were found not optimal for the purposes of operational hydrologic forecasting at Statkraft.

Originally aiming for incorporation in General Circulation Models, the Variable Infiltration Capacity (VIC) model (Liang et al., 1994; Hamman et al., 2018) has been used to address topics ranging from water resources management to land-atmosphere interactions and climate change. In the course of its development history of over 20 years, VIC has served as both a hydrologic model and land surface scheme. The VIC model is characterized by a grid-based representation of the model domain, statistical representation of sub-grid vegetation heterogeneity, and multiple soil layers with variable infiltration, and non-linear base flow. Inclusion of topography allows for orographic precipitation and temperature lapse rates. Adaptions of VIC allow the representation of water management effects and reservoir operation (Haddeland et al., 2006b, a, 2007). Routing effects are typically accounted for within a separate model during post-processing.

Directed towards the use in cold and seasonally snow covered small to medium sized basins, the Cold Regions Hydrological Model (CRHM) is a flexible object-oriented software system. CRHM provides a framework that allows the integration of physically-based parametrizations of hydrological processes. Current implementations consider cold region specific processes

such as blowing snow, snow interception in forest canopies, sublimation, snowmelt, infiltration into frozen soils, and hillslope water movement over permafrost (Pomeroy et al., 2007). CRHM supports both spatially-distributed and aggregated model approaches. Due to the object oriented structure, CRHM is used as both a research and predictive tool that allows rapid incorporation of new process algorithms. New and already existing implementations can be linked together to form a complete hydrological model. Model results can either be exported to a text file, ESRI ArcGIS, or a Microsoft Excel spreadsheet.

The Structure for Unifying Multiple Modelling Alternatives (SUMMA) (Clark et al., 2015a, b) is a hydrologic modelling approach that is characterized by a common set of conservation equations and a common numerical solver. SUMMA constitutes a framework that allows to test, apply and compare a wide range algorithmic alternatives for certain aspects of the hydrological cycle. Models can be applied to a range of spatial configurations (e.g., nested multi-scale grids and HRUs). By enabling model inter-comparison in a controlled setting, SUMMA is designed to explore the strengths and weaknesses of certain model approaches and provides a basis for future model development.

While all these models provide functionality similar to (and beyond) Shyft's model structure, such as flexibility in the computational subunit structure, allowing for using the multiple working hypothesis, and statistical representation of sub-grid landtype representation, the philosophy behind Shyft is fundamentally different from the existing model frameworks. These differences form the basis of the decision to develop a new framework, as outlined in the following section.

## 1.2 Why build a new hydrologic framework?

Given the abundance of hydrologic models and modelling systems, the question must be asked as to, why there is a need to develop a new framework. Shyft is a distributed modelling environment intended to provide operational forecasts for hydropower production. We include the capability of the exploration of multiple hydrologic model configurations, but the framework is somewhat more restricted and limited from other tools addressing the multiple model working hypothesis. As discussed in Section 1.1, several such software solutions exist, however, for different reasons these were found not suitable for deployment. The key criteria we sought when evaluating other software included:

- Open Source License and clear License Description

- Readily accessible software (e.g. not trial or registration based)

- High quality code

    well-commented

    modern standards

    api-based, not a GUI

    highly configurable using Object Oriented standards

- Well documented software

As we started the development of Shyft, we were unable to find a suitable alternative based on the existing packages at the time. In some cases the software is simply not readily available or suitably licensed. In others, documentation and test coverage was not sufficient. Most prior implementations of the multiple working hypothesis have a focus on the exploration of model uncertainty or provide more complexity than required, therefore adding data requirements. While Shyft provides some mechanisms for such investigation, we have further extended the paradigm to enable efficient evaluation of multiple forcing datasets, in addition to model configurations, as this is found to drive a significant component of the variability.

Notable complications arise in continuously operating environments. Current IT practices in the industry impose severe constraints upon any changes in the production systems, in order to ensure required availability and integrity. This challenges introduction of new modelling approaches, as service level and security are forcedly prioritised above innovation. To keep the pace of research, the operational requirements are embedded into automated testing of Shyft. Comprehensive unit test coverage provides proof for all levels of the implementation, whilst system and integration tests give objective means to validate the expected service behavior as a whole, including validation of known security considerations. Continuous integration aligned with agile (iterative) development cycle minimize human effort for the appropriate quality level. Thus, adoption of the modern practices balances tough IT demands with motivation for the rapid progress. Furthermore, C++ was chosen as a programming language for the core functionality. In spite of a steeper learning curve, templated code provides long term advantages for reflecting the target architecture in a sustainable way and the detailed documentation gives a comprehensive explanation of the possible entry-points for the new routines.

One of the key objectives was to create a well defined api, allowing for an interactive configuration and development from the command line. In order to provide the flexibility needed to address the variety of problems met in operational hydrologic forecasting, flexible design of workflows is critical. By providing a Python/C++ api, we provide access to Shyft functionality via the interpreted high-level programming language Python. This concept allows a Shyft user to design workflows by writing Python scripts rather than requiring user input via a graphical user interface (GUI). The latter is standard in many software products targeted toward hydropower forecasting, but was not desired. Shyft development is conducted by writing code in either Python or C++ and is readily scripted and configurable for conducting simulations programmatically.

## 2 Design principles

Shyft is a toolbox that has been purpose-developed for operational, regional-scale hydropower inflow forecasting. It was inspired from previous implementations of the multiple working hypothesis approach to provide the opportunity to explore multiple model realizations and gain insight into forecast uncertainty (Kolberg and Bruland, 2014; Clark et al., 2015b). However, key design decisions have been taken toward the requirement to provide a tool suitable for operational environments which vary from what may be prioritized in a pure research environment. In order to obtain the level of code quality and efficiency required for use in the hydropower market, we adhered to the following design principles:

## 2.1 Enterprise level software

Large organizations often have strict requirements regarding software security, testing, and code quality. Shyft follows the latest code standards and provides well documented source code. It is released as Open Source software and maintained on https://gitlab.com/shyft-os. All changes to the source code are tracked, and changes are run through a test suite greatly reducing the risk of errors in the code. This process is standard operation for software development, but remains less common for research software. Test coverage is maintained at greater than 90% of the whole c++ codebase. Python coverage is about 60% overall, including user-interface, which is difficult to test. Hydrology part has python test coverage more than 70% on average and is constantly validated via research activities.

## 2.2 Direct connection to data stores

A central philosophy of Shyft is that 'Data should live at the source!'. In operational environments, a natural challenge exists between providing a forecast as rapidly as possible, and conducting sufficient quality assurance and control (QA/QC). As the QA/QC process is often on-going, there may be changes to source datasets. For this reason, intermediate data files should be excluded, and Shyft is developed with this concept in mind. Users are encouraged to create their own 'repositories' that connect directly to their source data, regardless of the format (see Section 4).

## 2.3 Efficient integration of new knowledge

Research and Development (R&D) is critical for organizations to maintain competitive positions. There are two prevailing pathways for organizations to conduct R&D: through internal divisions or through external partnerships. The challenge of either of these approaches is that often the results from the research – or 'project deliveries' are difficult to implement efficiently in an existing framework. Shyft provides a robust operational hydrologic modelling environment, while providing flexible 'entry points' for novel algorithms, and the ability to test the algorithms in parallel to operational runs.

## 2.4 Flexible method application

Aligning with the principle of enabling rapid implementation of new knowledge, it is critical to develop a framework that enables flexible, exploratory research. The ability to quantify uncertainty is highly sought. One is able to explore epistemic uncertainty (Beven, 2006) introduced through the choice of hydrologic algorithm. Additionally, mechanisms are in place to enable selection of alternative forcing datasets (including point vs. distributed) and to explore variability resulting from these data.

## 2.5 Hot service

Perhaps the most ambitious principle is to develop a tool that may be implemented as a *hot service*. The concept is that rather than model results being saved to a database for later analysis and visualization, a practitioner may request simulation results for a certain region at a given time by running the model *on the fly* without writing results to file. Furthermore, perhaps one

would like to explore slight adjustments to some model parameters, requiring recomputation, in real time. This vision will only be actualized through the development of extremely fast and computationally efficient algorithms.

The adherence to a set of design principles creates a software framework that is consistently developed and easily integrated into environments requiring tested, well commented/documented, and secure code.

## 3    Architecture and Structure

Shyft is distributed in three separate code repositories and a 'docker' repository as described in Section 7.

Shyft utilizes two different codebases (see overview given in Figure 1). Basic data structures, hydrologic algorithms, and
models are defined in Shyft's core, which is written in C++ in order to provide high computational efficiency. In addition, an api exposes the data types defined in the core to Python. Model instantiation and configuration can therefore be utilized from pure Python code. In addition, Shyft provides functionalities that facilitate configuration and realization of hydrologic forecasts in operational environments. These functionalities are provided in Shyft's orchestration and are part of the Python codebase. As one of Shyft's design principles is that data should live at the source rather than Shyft requiring a certain input data format,
data repositories written in Python provide access to data sources. In order to provide robust software, automatic unit tests cover large parts of both codebases. In the following Section, details to each of the architectural constructs are given.

### 3.1    Core

The C++-core contains several separate code folders: core – for handling framework related functionality, like serialization and multithreading, timeseries – aimed for operating with generic timeseries and hydrology – all the hydrologic algorithms,
including structures and methods to manipulate with spatial information.[1] The design and implementation of models aims for multi-core operations, to ensure utilization of all computations resources available. At the same time, design considerations ensure the system may be run on multiple nodes. The core algorithms utilize third-party high performance, multithreaded libraries. These include the standard C++ (latest version), boost (Demming et al., 2010), armadillo (Sanderson and Curtin, 2016), and dlib (King, 2009) libraries, altogether leading to efficient code.
The Shyft-core itself is written using C++ templates from the above mentioned libraries, and also provides templated algorithms that consume template arguments as input parameters. The algorithms also return templates in some cases. This allows for high flexibility and simplicity, without sacrificing performance. In general, templates and static dispatch are used over class-hierarchies and inheritance. The goal toward faster algorithms is achieved via optimizing the composition and enabling multi-threading and the ability to scale out to multiple nodes.

---

[1]Core also contains dtss – time series handling services, energy_market – algorithms related to energy market modeling and web_api – web services, which are out of scope of this introductory paper.

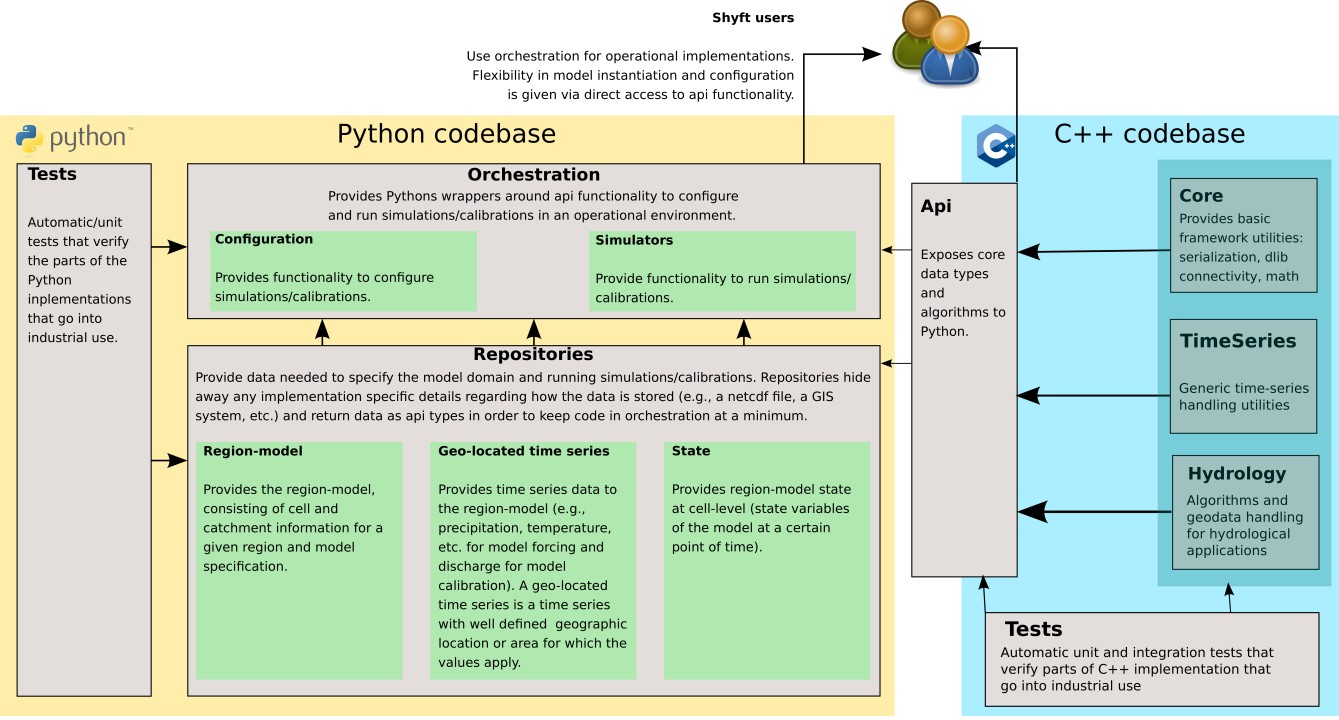

**Figure 1.** Architecture of Shyft.

## 3.2 Shyft api

The Shyft api exposes all relevant Shyft core implementations that are required to configure and utilize models to Python. The api is therefore the central part of the Shyft architecture that a Shyft user is encouraged to focus on. An overview of fundamental Shyft api types and how they can be used to initialize and apply a model is shown in Figure 2.

A user aiming to simulate hydrological models can do this by writing pure Python code without ever being exposed to the C++ codebase. Using Python, a user can configure and run a model, and access data at various levels such as model input variables, model parameters, and model state and output variables. It is of central importance to mention that as long as a model instance is initiated, all of this data is kept in the Random Access Memory of the computer, which allows a user to communicate with a Shyft model and its underlying data structures using an interactive Python command shell such as the Interactive Python (IPython) shell (Figure 3). In this manner, a user could for instance interactively configure a Shyft model, feed forcing data to it, run the model, and extract and plot result variables. Afterwards, as the model object is still instantiated in the interactive shell, a user could change the model configuration, e.g. by updating certain model parameters, re-run the model, and extract the updated model results. Exposing all relevant Shyft core types to an interpreted programming language, provides a considerable level of flexibility at the user level, that facilitates the realization of a large number of different operational setups. Furthermore, using Python offers a Shyft user the access to a programming language with intuitive and easy to learn syntax, wide support

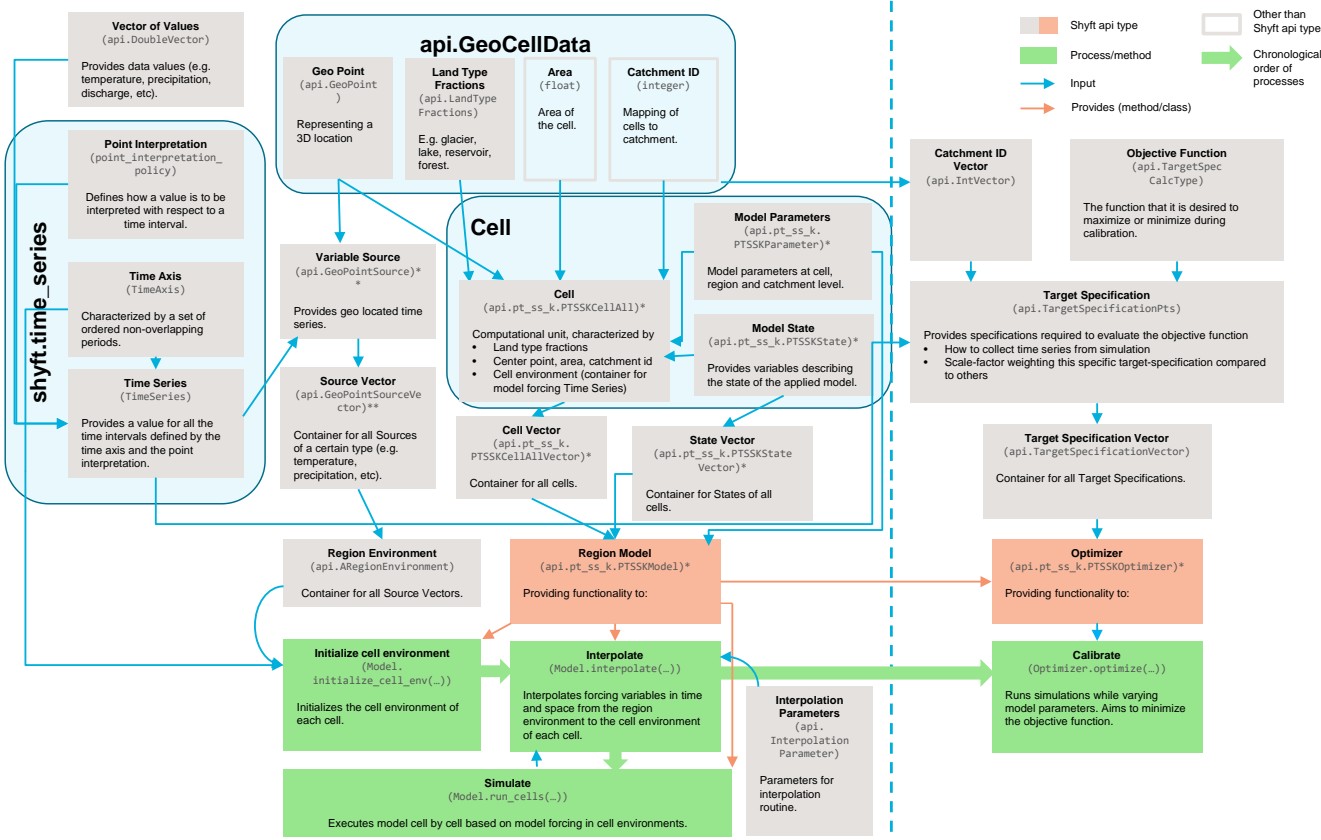

**Figure 2.** Description of the main Shyft api types and how they are used in order to construct a model. Api types used for running simulations are shown to the left of the dashed line, additional types used for model calibration to the right of it. * Different api types exist for different Shyft models dependent on the choice of the model. For this explanatory figure we use PTSSK, which is an acronym from Priestley-Taylor-Skaugen-Snow-Kirchner stack; ** Different api types exist for different types of input variables (e.g., temperature, precipitation, relative humidity, wind speed, radiation).

through a large and growing user community, over 300 standard library modules that contain modules and classes for a wide variety of programming tasks, and cross-platform availability.

All Shyft classes and methods available through the api follow the documentation standards introduced in the Guide to NumPy/SciPy Documentation [2]. Here we will try to give an overview of the types typically used in advanced simulations via api. (The comprehensive set of examples is available at https://gitlab.com/shyft-os/shyft-doc/tree/master/notebooks/api).

***shyft.time_series*** provides mathematical and statistical operations and functionality for time series. A time-series can be an expression, or a concrete point time-series. All time-series do have a time-axis (*TimeAxis* – a set of ordered non-overlapping periods), values (*api.DoubleVector*), and a point interpretation policy (*point_interpretation_policy*). The time-series can provide a value for all the intervals, and the point interpretation policy defines how the values should be interpreted: a) the point instant value is valid at the start of the period, linear between points, extend flat from last point to +∞, nan before first value;

---

[2]https://docs.scipy.org/doc/numpy-1.15.0/docs/howto_document.html

it is typical for state-variables, like water-level, temperature measured at 12:00 etc; b) the point average value represents an average or constant value over the period; it is typical for model-input and results, like precipitation and discharge. The Time-Series functionality includes resampling: average, accumulate, time_shift; statistics: min/max, correlation by nash-sutcliffe, kling-gupta; filtering: convolution, average, derivative; quality and correction: min-max limits, replace by linear interpolation or replacement time series; partitioning and percentiles.

*api.GeoCellData* represents common constant cell properties across several possible models and cell assemblies. The idea is that most of our algorithms use one or more of these properties, so we provide a common aspect that keeps this together. Currently it keeps the mid-point *api.GeoPoint*, the *Area*, *api.LandTypeFractions* (forest, lake, reservoir, glacier and unspecified), *Catchment ID* and routing information.

*Cell* is a container of GeoCellData and TimeSeries of model forcings (*api.GeoPointSource*). The Cell is also specific to the

*Model* selected, so *api.pt_ss_k.PTSSKCellAll* actually represents cells of a PTSSK-type, related to the *stack* selected (described in section 5.2). The structure collects all the necessary information, including cell state, cell parameters and simulation results. *Cell Vector* (*api.pt_ss_k.PTSSKCellAllVector*) is a container for the cells.

*Region Model* (*api.pt_ss_k.PTSSKModel*) contains all the *Cells* and also *Model Parameters* at region and catchment level (*api.pt_ss_k.PTSSKParameter*). Everything is vectorized, so, for example, *Model State* vector in the form of *api.pt_ss_k.-*

*PTSSKStateVector* collects together states of each model cell. Region Model is a provider of all functionality available: initialization (*Model.initialize_cell_env(...)*), interpolation (*Model.interpolate(...)*), simulation (*Model.run_cells(...)*) and calibration (*Optimizer.optimize(...)*, where optimizer is *api.pt_ss_k.PTSSKOptimizer* – also a construct within the model purposed specifically for the calibration. It is in the optimizer, where the *Target Specification* reside. To guide the model calibration we have a *GoalFunction*, that we try to minimize based on the *TargetSpecification*.

The Region Model is separated from **Region Environment** (*api.ARegionEnvironment*), which is a container for all *Sources* vectors of certain types, like temperature, precipitation etc in the form of *api.GeoPointSourceVector*.

Details on the main components of figure 2 are provided in following sections. Via api the user can interact with the system at any possible step, so the framework gives flexibility at any stage of simulation, but the implementation reside in C++ part keeping the efficiency at highest possible levels. Documentation page https://gitlab.com/shyft-os/shyft-doc/blob/master/

notebooks/shyft-intro-course-master/run_api_model.ipynb provides simple single-cell example of shyft simulation via api, which extensively explains each step.

## 3.3  Repositories

Data required to conduct simulations is dependent on the hydrological model selected. However, at present the available routines require at a minimum temperature and precipitation, and most also use wind speed, relative humidity and radiation.

More details regarding the requirements of these data are given in Section 5.

Shyft accesses data required to run simulations through *repositories* (Fowler, 2002). The use of respositories is driven by the aforementioned design principle to have a "direct connection to the data store". Each type of repository has a specific responsibility, a well defined interface, and may have a multitude of implementations of these interfaces. The data accessed by

```
[0]: # Initiating a model in Shyft

[1]: from shyft import api

[2]: model_type = api.pt_ss_k.PTSSKModel

[3]: cell = model_type.cell_t()

[4]: cell_vector = model_type.cell_t.vector_t()

[5]: cell_vector.append(cell)

[6]: region_parameter = model_type.parameter_t()

[7]: region_model = model_type(cell_vector, region_parameter)

...

[n]: region_model.run()
```

**Figure 3.** Simplified example showing how a Shyft user can configure a Shyft model using the Shyft api (`from shyft import api`) and (interactive) Python scripting. In line 2, the model to be used is chosen. In line 3 a model cell suitable to the model is initiated. In line 4 a cell vector, which acts as a container for all model cells, is initiated and the cell is appended to the vector (line 5). In line 6, a parameter object is initiated that provides default model parameters for the model domain. Based on the information contained in the cell vector (defining the model domain), the model parameters, and the model itself, the region-model can be initiated (line 7) and, after some intermediate steps not shown in this example, stepped forward in time (line n). The example is simplified in that it gives a rough overview on how to use the Shyft api, but does not provide a real working example. The functionality shown herein provides a small subset of the functionalities provided by the Shyft api. For more complete examples we recommend the Shyft documentation (https://shyft.readthedocs.io).

repositories usually origin from a relational database or file-formats that are well known. In practice, data is never accessed in any other way than through these interfaces, and the intention is that data is never converted into a particular format for Shyft. In order to keep code in the Shyft orchestration at a minimum, repositories are obliged to return Shyft api types. Shyft provides interfaces for the following repositories:

**Region-model repository**

> The responsibility is to provide a configured region-model, hiding away any implementation specific details regarding how the model configuration and data is stored (e.g., in a netcdf database, a GIS-system, etc.).

**Geo-located time-series repository**

> The responsibility is to provide all meteorology and hydrology relevant types of geo-located time-series needed to run or calibrate the region-model (e.g., data from station observations, weather forecasts, climate models, etc.).

**Interpolation parameter repository**

The responsibility is to provide parameters for the interpolation method used in the simulation.

**State repository**

The responsibility is to provide model states for the region-model and store model states for later use.

Shyft provides implementations of the region-model repository interface and the geo-located time series repository interface for several datasets available in netcdf formats. These are mostly used for documentation and testing and can likewise be

utilized by a Shyft user. Users aiming for an operational implementation of Shyft are encouraged to write their own repositories following the provided interfaces/examples rather than converting data to the expectations of the provided netcdf repositories.

## 3.4   Orchestration

We define 'Orchestration' as the composition of the simulation configuration. This included defining the model domain, selection of forcing datasets and model algorithms, and presentation of the results. In order to facilitate the realization of simple

hydrologic simulation and calibration tasks, Shyft provides an additional layer of Python code. The Shyft orchestration layer is built on top of the api functionalities and provides a collection of utilities that allow to configure, run, and post-process simulations. Orchestration provides for two main objectives.

Firstly, to offer an easy entry point for modellers seeking to use Shyft. By using the orchestration, users require only a minimum of Python scripting experience in order to configure and run simulations. However, the Shyft orchestration gives

only limited functionality and users might find it limiting their ambitions. For this reason, Shyft users are strongly encouraged to learn how to effectively use Shyft api functionality in order to be able to enjoy the full spectrum of opportunities that the Shyft framework offers for hydrologic modelling.

Secondly, and importantly, it is through the orchestration that full functionality can be utilized in operational environments. However, as different operational environments have different objectives, it is likely that an operator of an operational service

wants to extend the current functionalities of the orchestration or design a completely new one from scratch suitable to the needs the operator defines. The orchestration provided in Shyft then rather serves as an introductory example.

## 4   Conceptual Model

The design principles of Shyft led to the development of a framework that attempts to strictly separate the model domain (*region*) from the model forcing data (*region environment*) and the model algorithms in order to provide a high degree of

flexibility in the choice of each of these three elements. In this Section it is described how a model domain is constructed in Shyft, and how it is combined with a set of meteorological forcing data and a hydrological algorithm in order to generate an object that is central to Shyft, the so called *region-model*. For corresponding Shyft api types see Figure 2.

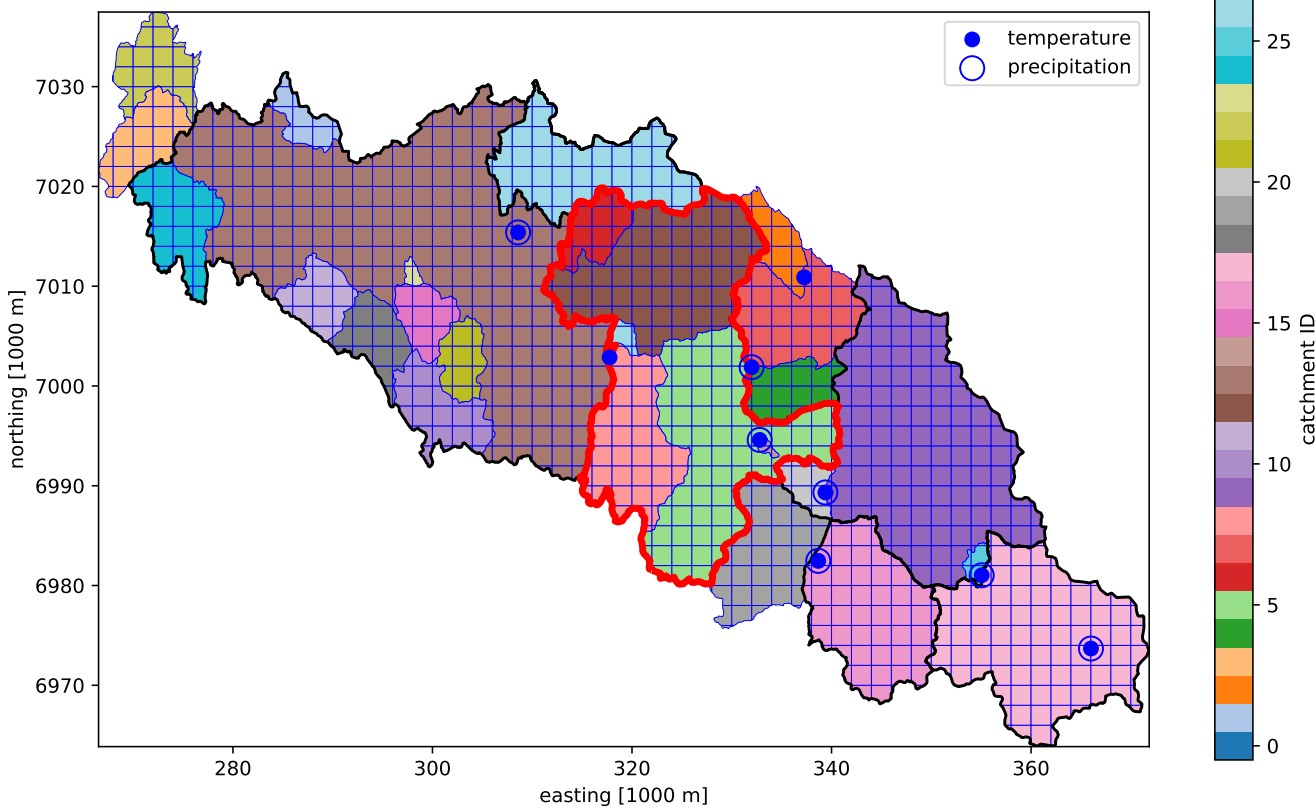

**Figure 4.** A Shyft model domain existing of a collection of cells. Each cell is mapped to a catchment using a catchment id. Default cell shape in this example is square, however, note that at the boundaries cells are not square but instead follow the basin boundary polygon. The red line indicates a catchment that could be defined by a sub-set of catchment ids. The framework would allow for using the full region, but simulating only within this catchment. The blue circles mark the geographical location of meteorological data sources, which are provided by the region environment.

## 4.1  Region: the model domain

In Shyft, a model domain is defined by a collection of geo-located sub-units called *cells*. Each cell has certain properties such as land type fractions, area, geographic location, and a unique identifier specifying to which catchment the cell belongs (the catchment id). Cells with the same catchment id are assigned to the same catchment and each catchment is defined by a set of catchment ids (see Figure 4). The Shyft model domain is composed of a user defined number of cells and catchments, and is called a *region*. A Shyft region thus specifies the geographical properties required in a hydrologic simulation.

For computations, the cells are vectorized rather than represented on a grid, as is typical for spatially distributed models. This aspect of Shyft provides significant flexibility and efficiency in computation.

## 4.2 Region environment

Model forcing data is organized in a construct called *region environment*. The region environment provides containers for each variable type required as input to a model. Meteorological forcing variables currently supported are temperature, precipitation, radiation, relative humidity and wind speed. Each variable container can be fed a collection of geo-located time series, referred to as *sources*, each providing the timeseries data for the variables coupled with methods that provide information about the geographical location for which the data is valid. The collections of sources in the region environment can originate from e.g. station observations, gridded observations, gridded numerical weather forecasts, or climate simulations (see Figure 4). The time series of these sources are usually presented in the original time resolution as available in the database from which they originate. That is, the region environment typically provides meteorological raw data, with no assumption on spatial properties of the model cells or the model time step used for simulation.

## 4.3 Model

The model approach used to simulate hydrological processes is defined by the user and independent from the choice of the region and region environment configurations. In Shyft, a model defines a sequence of algorithms, each of which describing a method to represent certain processes of the hydrological cycle. Such processes might be evapotranspiration, snow accumulation and melt processes, or soil response. The respective algorithms are compiled into *model stacks*, where different model stacks differ in at least one method. Currently, Shyft provides four different model stacks described in more detail in Section 5.2.

## 4.4 Region-model

Once a user has defined the region representing the model domain, the region environment providing the meteorological model forcing, and the model defining the algorithmic representation of hydrologic processes, these three objects can be combined to create a *region-model*, an object that is central to Shyft.

The *region-model* provides the following key functionalities that allow to simulate the hydrology of a region:

- Interpolation of meteorological forcing data from the source locations to the cells using a user defined interpolation method, and interpolation from the source time resolution to the simulation time resolution. A construct named *cell environment*, a property of each cell, acts as container for the interpolated time series of forcing variables. Available interpolation routines are descried in Section 5.1.

- Running the model forward in time. Once the interpolation step is performed, the region-model is provided with all data required to predict the temporal evolution of hydrologic variables. This step is done through cell-by-cell execution of the model stack. This step is computationally highly efficient due to enabled multithreading that allows parallel execution on a multiprocessing system, utilizing all Central Processing Units (CPU) unless otherwise specified.

- Providing access to all data related to region and model. All data that is required as input to the model and generated during a model run is stored in memory and can be accessed through the region-model. This applies to model forcing data

at source and cell level, model parameters at region and catchment level, static cell data, and time series of model state result variables. The latter two are not necessarily stored by default in order to achieve high computational efficiency, but collection of those can be enabled prior to a model run.

A simplified example how to use the Shyft api to configure a Shyft region-model is shown in Figure 3 or one can consult documentation: https://gitlab.com/shyft-os/shyft-doc/blob/master/notebooks/shyft-intro-course-master/run_api_model.ipynb

## 4.5 Targets

Shyft provides functionality to estimate model parameters by providing implementation of several optimization algorithms and goal functions. Shyft utilizes optimization algorithms from **dlib**: www.dlib.net/optimization.html#find_min_bobyqa –
Bound Optimization BY Quadratic Approximation (BOBYQA), which is a derivative-free optimization algorithm, explained in (Powell, 2009) and global function search algorithm http://dlib.net/optimization.html#global_function_search, which performs global optimization of a function, subject to bounds constrains.

     In order to optimize model parameters, model results are evaluated against one or several target specifications (Gupta et al., 1998). Most commonly, simulated discharge is evaluated with observed discharge, however, Shyft supports further variables
such as mean catchment SWE or snow-covered area (SCA) to estimate model parameters. This enables refined condition of the parameter set for variables for which a more physical model may be used and high quality data is available. This approach is being increasingly employed in snow-dominated catchments (e.g. Teweldebrhan et al. (2018b); Riboust et al. (2019)). An arbitrary number of target time series can be evaluated during a calibration run, each representing a different part of the region and/or time interval and step. The overall evaluation metric is calculated from a weighted average of the metric of each target
specification. To evaluate performance user can specify Nash-Sutcliffe (Nash and Sutcliffe, 1970), Kling-Gupta (Gupta et al., 1998), or Absolute Difference or Root Mean Square Error (RMSE) functions. The user can specify which model parameters to optimize, giving a search range for each of the parameters. In order to provide maximum speed, the optimized models are used during calibration, so that the CPU and memory footprints are minimal.

## 5   Hydrologic modelling

Modelling the hydrology of a region with Shyft is typically done by first interpolating the model forcing data from the source locations (e.g. atmospheric model grid points or weather stations) to the Shyft cell location and then running a model stack cell-by-cell. This Section gives an overview over the methods implemented for interpolation and hydrologic modelling.

### 5.1   Interpolation

In order to interpolate model forcing data from the source locations to the cell locations, Shyft provides two different interpo-
lation algorithms: interpolation via inverse distance weighting and Bayesian Kriging. However, it is important to mention that Shyft users are not forced to use the internally provided interpolation methods. Instead, the provided interpolation step can be

skipped and input data can be fed directly to cells, leaving it up to the Shyft user how to interpolate/downscale model input data from source locations to the cell domain.

### 5.1.1 Inverse distance weighting

Inverse Distance Weighting (IDW) (Shepard, 1968) is the primary method used to distribute model forcing timeseries to the cells. The implementation of IDW allows a high degree of flexibility in the approach of a choice of models for different variables.

### 5.1.2 Bayesian temperature kriging

As described in section 5.1.1 we provide functionality to use a height-gradient based approach to reduce the systematic error
when estimating the local air temperature based on regional observations. The gradient value may either be calculated from the data or set manually by the user.

In many cases, this simplistic approach is suitable for the purposes of optimizing the calibration. However, if one is interested in greater physical constraints on the simulation, we recognize the gradient is often more complicated and varies both seasonally and with local weather. There may be situations in which insufficient observations are available to properly calculate the
temperature gradient, or potentially the local forcing at the observation stations are actually representative of entirely different processes than the one for which the temperature is being estimated. An alternative approach has therefore been implemented in Shyft, that enables applying a method that would buffer the most severe local effects in such cases.

The application of Bayes' Theorem is suitable for such weighting of measurement data against prior information. Shyft provides a method that estimates a regional height gradient and sea level temperature for the entire region, which together with
elevation data subsequently model a surface temperature.

### 5.1.3 Generalization

The IDW in shyft is generalized, and adapted to the practicalities using available grid forecasts:

1. Selecting the neighbours, that should participate in the IDW, individually, for each destination point.

   - The Z-scale allows the selection to discriminate neighbors that are of different height.(e.g. precipitation, relative
humidity, prefer same heights)

   - Number of neighbors that should be chosen for any given interpolation point,

   - Excluding neighbors with distances larger than specified limit

2. Given the neighbors selected according to (1): A transformation technique/method adapted to the signal type is applied, to project the signal from it's source position into the destination position. The weight scaling factor is $1/pow(distance, distance\_scale$

- temperature has several options available:

| Input variable | Unit | Model Stacks |
|---|---|---|
| Temperature | °C | all model stacks |
| Precipitation | mm hr$^{-1}$ | all model stacks |
| Radiation | W m$^{-2}$ | all model stacks |
| Wind speed | m s$^{-1}$ | PTGSK |
| Relative humidity | % | PTGSK |

**Table 1.** Input data requirements per model.

    – The temperature lapse-rate is computed using the nearest neighbors with sufficient/maximized vertical distance

    – The full 3d temperature flux vector is derived from the selected points, and then the vertical component is used.

- precipitation: $(scale\_factor)^{(z-distance-in-meters/100.0)}$, scale-factor specified in parameter, z-distance as source-destination distance

- radiation: allows slope/factor adjustment on the destination cell

## 5.2 Model stacks

In Shyft, a hydrologic model is a sequence of hydrologic methods and called *model stack*. Each method of the model stack describes a certain hydrologic process and the model stack typically provides a complete rainfall-runoff model. In the current state, the model stacks provided in Shyft differ mostly in the representation of snow accumulation and melt processes due to the predominant importance of snow in the hydropower production environments of Nordic countries, where the model has been operationalized first. These model stacks provide sufficient performance in the catchments for which the model has been evaluated, however, it is expected that for some environments with different climatic conditions more advanced hydrologic routines will be required and therefore new model stacks are in active development. Furthermore, applying Shyft in renewable energy production environments other than hydropower (e.g. wind power) is realizable but will not be discussed herein.

Currently, there are four model stacks available that undergo permanent development. With the exception of the HBV Lindström et al. (1997) model stack, the distinction for the remaining three model options are the snow routines used in the hydrologic calculations. In these remaining model stacks, the model stack naming convention informs about the hydrologic methods used in the respective model.

## 5.3 PTGSK

- **PT** (**P**riestly-**T**ailor)

Method for evapotranspiration calculations according to Priestley and Taylor (1972).

- **GS** (**G**amma-**S**now)

  Energy balance based snow routine that uses a gamma function to represent sub-cell snow distribution (Kolberg et al., 2006).

- **K** (**K**irchner)

  Hydrologic response routine based on Kirchner (2009).

In the PTGSK model stack, the model first uses Priestley-Taylor to calculate the potential evapotranspiration based on temperature, radiation, and relative humidity data (see table 1 for an overview of model input data). The calculated potential evaporation is then used to estimate the actual evapotranspiration using a simple scaling approach. The Gamma-Snow routine is used to calculate snow accumulation and melt adjusted runoff using time series data for precipitation and wind speed in addition to the input data used in the Priestley-Taylor method. Glacier melt is accounted for using a simple temperature index approach (Hock, 2003). Based on the snow and ice adjusted available liquid water, Kirchner's approach is used to calculate the catchment response. The PTGSK model stack is the only model in Shyft which provides an energy-balance approach to the calculation of snow accumulation and melt processes.

## 5.4  PTSSK

- **SS** (**S**kaugen **S**now)

  Temperature index model based snow routine with focus on snow distribution according to Skaugen and Randen (2013) and Skaugen and Weltzien (2016).

As with the PTGSK model stack, all calculations are identical with the exception that the snow accumulation and melt processes are calculated using the Skaugen Snow routine. The implementation strictly separates potential melt calculations from snow distribution calculations, making it an easy task to replace the simple temperature index model currently in use with an advanced (energy balance based) algorithm.

## 5.5  PTHSK

- **HS** (**H**BV **S**now)

  Temperature index model for snow accumulation and melt processes based on the snow routine of the HBV (Hydrologiska Byråns Vattenbalansavdeling) model (Lindström et al., 1997).

As with the PTGSK model stack, all calculations are identical with the exception that the snow accumulation and melt processes are calculated using the snow routine from the HBV model.

## 5.6  HBV

The HBV model stack very closely resembles the original description of Bergström (1976). An exception is that we calculate the potential evapotranspiration using the Priestley-Taylor routine rather than temperature adjusted monthly mean potential

evapotranspiration. In the HBV model stack, the original routines are all combined into a complete model. As with the other routines, we also include the calculation of glacial melt and allow for routing using the methods described in Section 5.7.

## 5.7  Routing

Routing in Shyft is established through two phases: a) cell-to-river routing, and b) river network routing. In cell-to-river routing water is routed to the closest river object providing lateral inflow to the river. While individual cells have the possibility to have a specific routing velocity and distance, unit hydrograph (UHG) shape parameters are catchment specific. River network routing provides for routing from one river object to the next downstream river object along with lateral inflow from the cells as defined in the first phase. The sum of the upstream river discharge and lateral inflow is then passed to the next downstream river object. A UHG parameter structure provides for UHG shape parameters and a discretized time-length according to the model time-step resolution. Currently, a gamma function is used for defining the shape of the UHG. The approach of Skaugen and Onof (2014) to sum together all cell-responses at a river routing point and define a UHG based on a distance distribution profile to that routing point is commonly used. Together with convolution, the UHG will determine the response from the cells to the routing point.

## 5.8  Uncertainty analysis

Shyft is equipped with several mechanisms, which ease uncertainty analysis of different kinds. First, the modules are easily configurable via yaml configuration files (example of model configuration file is provided in listing 1), which are utilized by orchestration routines. The configuration files define:

- forcings datasets,

- interpolation methods,

- calibration and simulation periods,

- parameters to be used in calibrations,

- model stack to be used.

Secondly, via python api interface practitioner can interact with forcings, parameters, state variables at any stage of simulation, so in case the orchestration provided by Shyft is limited, one can programmatically control and manipulate simulations.

Thus, one can asses uncertainty coming from forcing data via model runs with variety of forcing datasets and same configuration (stack and parameters), uncertainty coming from model structure via running experiment with different stacks, uncertainty coming from parametrization of the stacks. All types of such experiments are possible without recompilation of the software. The uncertainty analysis application is presented in Teweldebrhan et al. (2018c) and an application of Shyft in a Machine Learning based environment is presented in Teweldebrhan et al. (2020).

## 5.9 Prediction in Ungauged Basins

Prediction in Ungauged basins (PUB) is mostly done via various methods under regionalization approach (Hrachowitz et al., 2013), where regionalization means that hydrological information from gauged (donor) basin is transferred to ungauged (target) location. Shyft is perfectly suited with functionality for parameters regionalization: given several catchments in the area, one can easily set up calibration procedure (via yaml configuration files or programmatically) to use one or some of the sub-catchments in the domain and for the remaining ungauged catchments. What provided immense flexibility here is the use of 'repositories', whereby a 'parameter' repository could be developed that maps gauged to ungauged catchments and returns the appropriate configuration. One could even take the approach of making the configuration functional, and apply gradient calculations to physical parameters (e.g. PCORR). The internal methods, such as inverse distance weighting (IDW) can be applied in order to average donor subcatchments results based on geographical proximity to the target (Merz and Blöschl, 2004). However, considerations on the similarities of the subcatchments and suitable regionalization approach are on the side of the practitioner.

## 5.10 Hydrological Forecasting

The development of Shyft was primarily driven by requirements within a production planning environment. Key metrics for the software were the ability to produce simulations and forecasts in a fast and accurate manner, while allowing for replicable configurations and assessment of uncertainty. In order to balance between optimal forecast performance and computational efficiency, several choices were made within domains for which the greatest control exists. For example, Shyft can run, in a highly optimized manner, simulations across numerous spatial domains, with different 'model stacks' and/or weather forcings. The feature-rich and programmatic configurability provides the "fast" capability.

"Accurate" means that all algorithmic implementations and hydrologic paradigms are coded in such a manner as to assure replicability and to optimally serve the quality of the discharge forecast, and only that. We do not place emphasis on "accuracy" in the context of simulating the full hydrologic system or state, but rather discharge as measured against observations. There are opinionated choices made in the currently available model stacks. For instance, given the uncertainties associated with the input data required to physically resolve subsurface flow, we intentionally leave the 'response' of the watershed to the Kirchner routine. However, we have full control over distributed input forcing data and surface geography, therefore we emphasize the development of snow distribution routines and weather processing routines to provide our optimal forecasts. So it should not be construed that within Shyft there is not an interest in the physical systems. Rather the contrary, but that emphasis is placed based on greatest value toward the goals of production planning.

The introduction of Shyft to the open-source community is an invitation for further development and we encourage the contribution of routines to address components of the hydrologic cycle for which contributors feel they can provide greater quality forecasts through implementation of more sophisticated routines.

## 6  Computational Performance

There are certain design principals, that indirectly prove Shyft is an efficient tool from computational point of view:

- The choice of programming language and librarires: C++, fast 3rd party libraries, boost, armadillo,dlib, template, static-dispatch, inline, simple structures;

- Avoiding memory-allocations during computation, traffic, shared-pointer, and other interlocking features in the core;

- Design of data-structures/types to maximize performance, e.g. time-series, fixed/or known-timestep time-axis, pre-allocate cell environment;

- Design of the computational steps (like first preparation/interpolation, then cells, then river-network);

- Design of hydrology methods that is suitable for the above approach, like composable method-stacks.

The formal comparison to other software is challenged by several things. First of all, most *operational tools* continue to reside as *proprietary*, meaning they are not available for general public. The most of the ready to use research tools are developing with the *research activities* as being the main driving factor, not taking into account production specifics described
in sections 1.2 paragraph 2. Another challenge comes from well-known problem of *standardized hydrological benchmarking* discussed in Abramowitz (2012) and Grewe et al. (2012), as the definition of efficient (from hydrological simulations point of view) software depends on the proper application of metrics and proper justification at each step.

However, we can asses computational efficiency via benchmarking towards existing hardware. Figure 5 shows results of the synthetic experiment on evaluating computational performance. As can be seen the use of computational time increase linearly,
which is the expected behavior.

## 7  Availability and documentation

The source code of Shyft is published under version 3 of the GNU Lesser General Public License. All code is available via git repositories located at: https://gitlab.com/shyft-os. Therein, three separate repositories are used by the developers for the management of the code. The main code repository is simply called 'shyft'. In addition to the source code of Shyft, data needed
to run the full test suite is distributed in the 'shyft-data' repository, while a collection of Jupyter Notebooks providing example Python code for a number of use cases of the Shyft api is provided within the 'shyft-doc' repository – in addition to the full code for the *Read the Docs* (https://shyft.readthedocs.io) website. At this site we provide end-user documentation on:

- installation on both Linux and Windows operating systems

- how to use the Shyft api to construct a hydrological model, feed it with data, and run hydrologic simulations

- use of Shyft repositories to access model input data and parameters

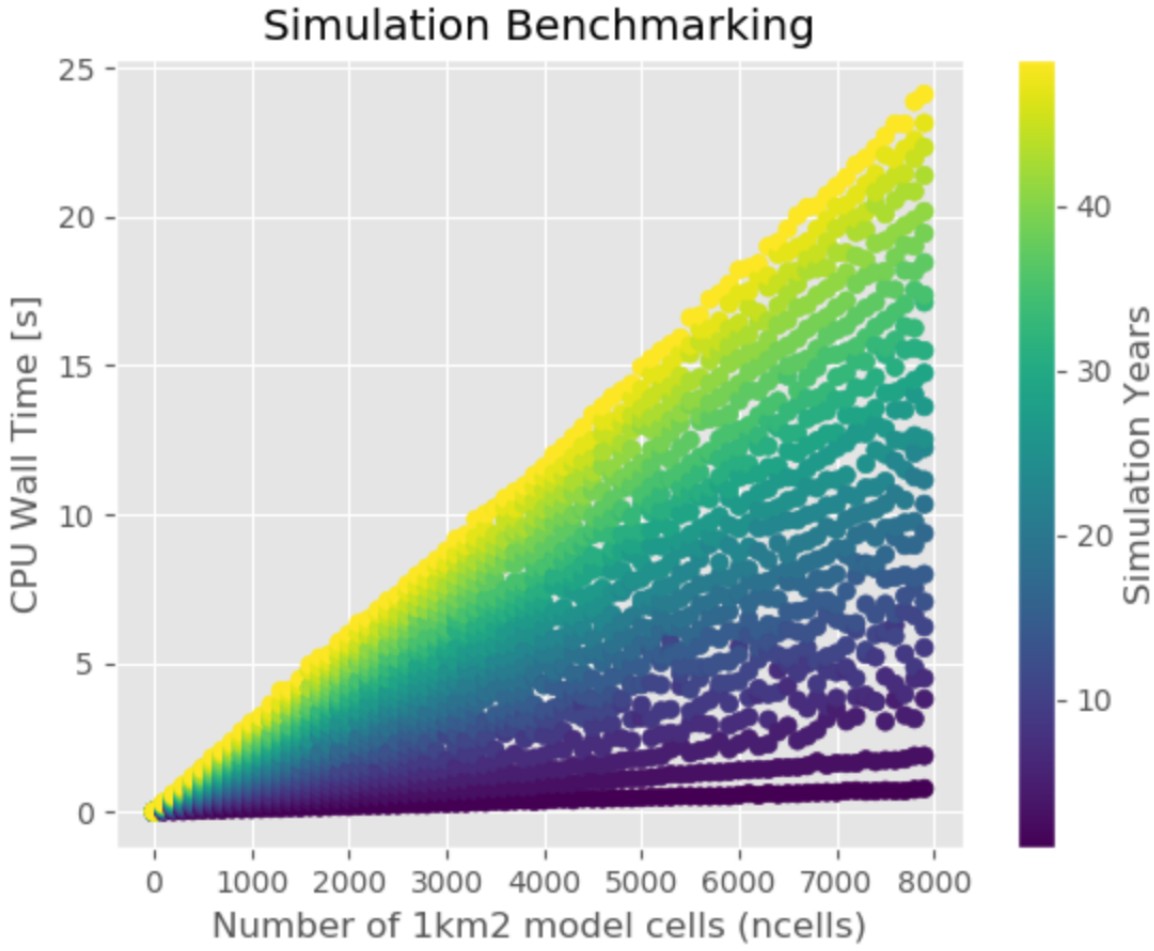

**Figure 5.** Benchmarking of Shyft computational performance conducted using a synthetic timeseries and regular 1km grid cells. The model was run at an hourly timestep for the purposes of testing with 3-hourly input data from the NOAA Global Ensemble Forecast System (GEFS).

- use of the Shyft orchestration to configure and run simulations

We also maintain 'dockers' repository https://gitlab.com/shyft-os/dockers, where docker build recipes for the complete Shyft eco-system reside, including 'build dockers', 'python test dockers', 'development dockers' and 'application dockers'.

An aspect of Shyft that is unique to most research codebases is the extensive test suite that covers both the Python and the C++ codebase. The test suite is comprehensive, and in addition to unit-tests covering C++ parts and Python parts, it also covers integration tests assuring valid bindings to external dependencies such as netcdf and geo-services. This is a particularly helpful resource for those who are somewhat more advanced in their knowledge of coding.

## 8 Recent Applications

Shyft was originally developed in order to explore epistemic uncertainty associated with model formulation and input selection (Beven et al., 2011). At Statkraft, and at most Norwegian hydropower companies, inflow forecasting to the reservoirs is conducted using the well-known HBV (Bergström, 1976) model. The inflow to the reservoirs is a critical variable in production planning. As such, there was an interest to evaluate and assess whether improvements in the forecasts could be gained through the use of different formulations. In particular, we sought the ability to ingest distributed meteorological inputs and to also assess the variability resulting from NWP models of differing resolution and operating at different time-scales (e.g. Zsoter et al. (2015))

### 8.1 Production Planning

In Figure 6 we present a simple example in which Shyft is used to provide inflow forecasts with a horizon of 15 days for a subcatchment in the Nea-Nidelva basin (marked red in Figure 4). The total area of the basin is about 3050 km$^2$ and the watercourse runs for some 160 km from the Sylan mountains on the boarder between Sweden and Norway to the river mouth in Trondheimsfjorden. The hydrology of the area is dominated by snow melt from seasonal snow cover. The intent of the example is not to analyze the performance of the forecast, but rather to simply demonstrate the capability of Shyft to run an ensemble forecast in a single configuration.

In this example we show the results from a single 'ensemble' configuration of Shyft in which the region is configured with a spatial resolution of 1x1 km$^2$ and the model setup aims to reproduce the hydrological forecast with forecasting start on 22.04.2018, 00:00 UTC. In order to estimate model state variables, the simulation initiates before the melt season begins. Using the model state based on the historical simulation and latest discharge observations, the model state is updated so that the discharge at forecast start equals the observed discharge. Forecasts are then initiated based on the updated model state and using a number of weather forecasts from different meteorological forecast providers and ensembles. A deterministic hydrologic forecast is run using the AROME weather prediction system from the Norwegian Meteorological Institute with a horizon of 66 hours and a spatial resolution of 2.5 km (Seity et al., 2011; Bengtsson et al., 2017). Likewise, a second deterministic forecast is conducted based on the high resolution 10-day forecast product from the European Centre for Medium-Range Weather Forecasts (ECMWF) (spatial resolution 0.1° x 0.1° latitude/longitude). In addition to the deterministic forecasts, simulations based on ECMWF's 15-day ensemble forecast system are conducted (51 ensemble members, spatial resolution 0.2° x 0.2° latitude/longitude) (Andersson and Thépaut, 2008). In total, from initiation to completion this forecast takes less a few minutes to run, with the bulk of the time dependent on the Input-Output bound operations which depends significantly on how the user chooses to implement their repositories for the weather (e.g. Section 3.3).

The forecast is run during the initial phase of the snow melt season in April 2018. The historical simulation overestimated streamflow during the week prior to the forecast start (left of black bar in Figure 6). However, after updating the model state using observed discharge (the black bar is the time-step, when the internal states updated to match observations), the simulations provide a reasonable streamflow forecast (right of black bar in Figure 6) as well as a series of possible outcomes

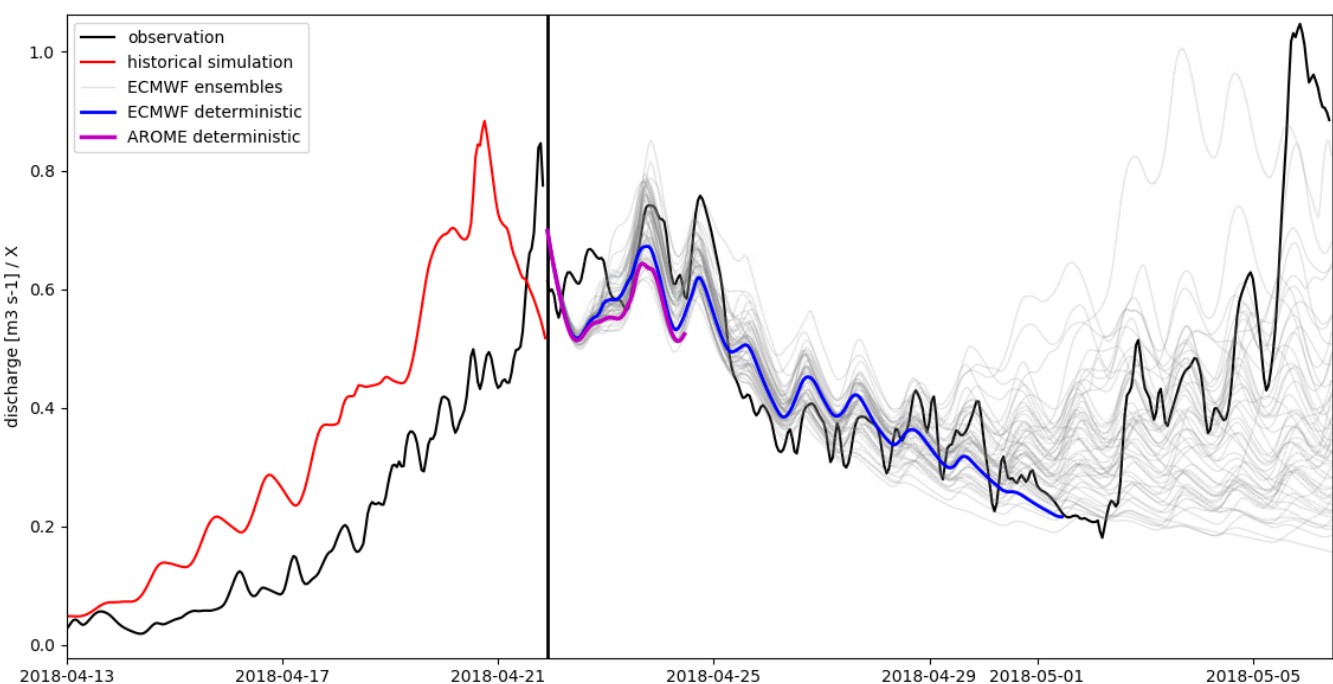

**Figure 6.** Example of a hydrological forecast conducted with Shyft for a subcatchment in the Nea-Nidelva basin. Shyft has been used to simulate the historical discharge (red) in order to estimate state variables at nowtime (22.04.2018, 00:00 UTC). Three different weather forecast products are then used in order to predict discharge: the operational deterministic weather prediction AROME with a forecasting horizon of 66 hours from the Norwegian Meteorological Institute (purple), ECMWF's deterministic weather forecast with a horizon of 10 days (blue), and ECMWF's ensemble weather prediction with a horizon of 15 days and 51 ensemble members (grey). Discharge observations (black) are shown for reference. Note: discharge are provided from Statkraft AS and are divided by a factor $X$ in order to mask the observational data as Statkraft's data policy considers discharge sensitive data.

based on the ensemble of meteorological products. For production planning purposes, the ability to assess the uncertainty of the forecast rapidly, and to efficiently ingest ensemble forecasts is highly valued (e.g. (Anghileri et al., 2016, 2019))).

## 8.2 The impact of aerosol-driven snowpack melt on discharge

One of the first research-based applications of the framework was to evaluate the impact of aerosols on snow melt. The story of 'Light-Absorbing Impurities in Snow and Ice' (LAISI), is one that has gained a signficant amount of attention in the research community. While the initial emphasis on the development of the routine was aligned more toward Arctic black carbon aerosol, the increasing awareness of the role of aerosols influencing discharge in regions of Colorado (Bryant et al., 2013) creates an exciting new application for this algorithm. To our knowledge no other catchment scale hydrologic forecast model provides this capability.

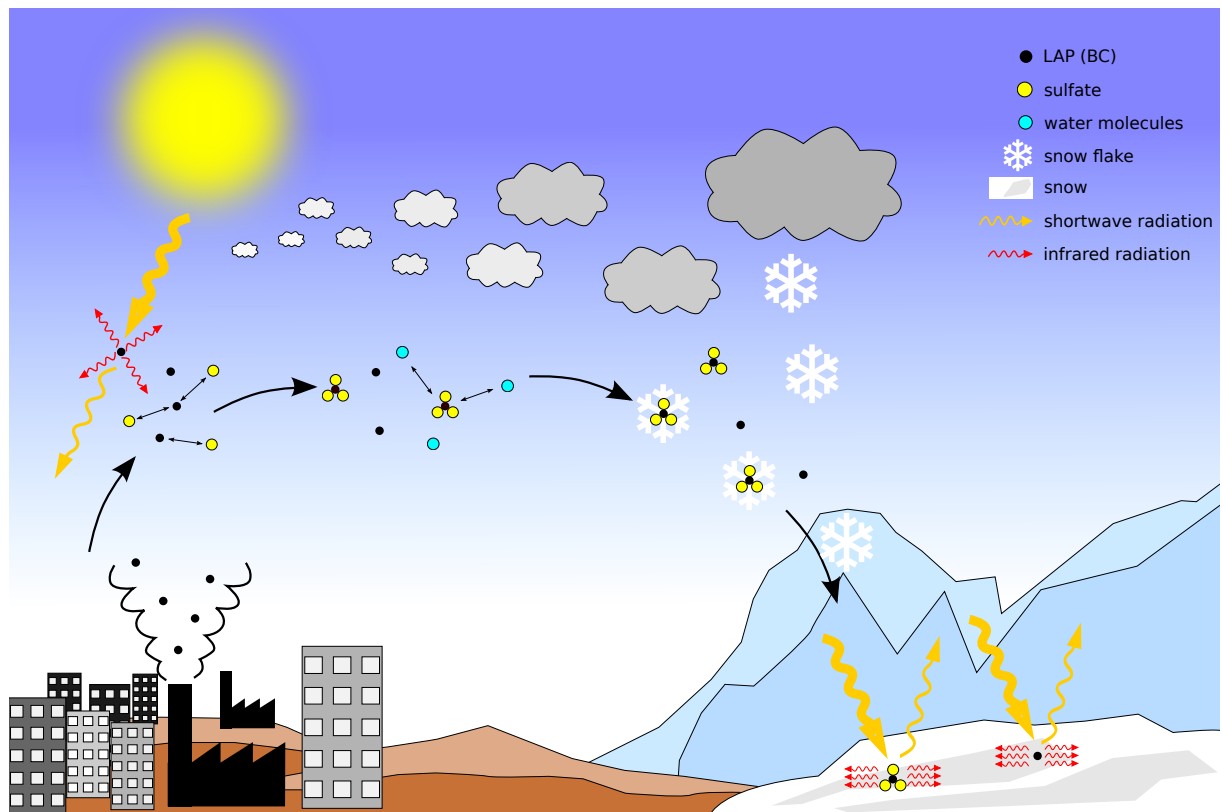

**Figure 7.** Schematic drawing of Black Carbon (and other light-absorbing particles) pathways and processes in atmosphere and snow.

Wiscombe and Warren (1980) and Warren and Wiscombe (1980) hypothesized that trace amounts of absorptive impurities occurring in natural snow can lead to significant implications for snow albedo. To date, many studies have given evidence to this hypothesis (Jacobson, 2004; IPCC, 2013; Wang et al., 2013; Hansen and Nazarenko, 2004, e.g.,). Particles that have the ability to absorb electromagnetic waves in the short wavelength range caught the attention of the research community due their influence on water and energy budgets of both the atmosphere and the earth surface (Twomey et al., 1984; Albrecht, 1989; Hansen et al., 1997; Ramanathan et al., 2001, e.g.,). If these aerosols are deposited alongside snowfall, they lower the spectral albedo of the snow in the shortwave spectrum (see for example Figure 1 of Hadley and Kirchstetter (2012)), and act in a similar way as their airborne counterparts by emitting infrared radiation (Figure 7). Due to the efficient absorption properties of snow grains in the thermal infrared, this leads to heating the snow. This in turn has implications for the evolution of the snow micro-structure (Flanner et al., 2007) and snow melt.

At the catchment scale such an absorptive process should have an observable impact on melt rates and discharge. While several studies have provided field-based measurements of the impact of LAISI on albedo of the snow (Painter et al., 2012; Doherty et al., 2016, e.g.,), no studies have attempted to address the impact of this process on discharge. Skiles and Painter

(2016) showed that snow pack melt rates were impacted in the Colorado Rockies resulting from dust deposition by evaluating a sequence of snow pits and Painter et al. (2017) provided observational evidence that the hydrology of catchments is likely impacted by LAISI deposition, but no studies incorporated a physically based simulation in a hydrologic model to ascribe the component of melt attributable to LAISI.

Using Shyft, Matt et al. (2018) addressed this process by using the catchment as an 'intergrating sampler' to capture the signal of deposited LAISI. In this work, it was shown that even in a remote Norwegian catchment, the timing of melt is impacted by the slight BC concentrations deposited in the snow with an average percentage increase in daily discharge ranging from 2.5 to 21.4 % for the early melt season and a decrease in discharge of -0.8 to -6.7 % during the late melt season, depending on the deposition scenario.

To accomplish this, a new snow pack algorithm was developed to solve the energy balance for incoming shortwave radiation flux $K_{in}$, incoming and outgoing longwave radiation fluxes $L_{in}$ and $L_{out}$, sensible and latent heat fluxes $H_s$ and $H_l$, and the heat contribution from rain $R$. As such, $\frac{\delta F}{\delta t}$ is the net energy flux for the snowpack:

$$\frac{\delta F}{\delta t} = K_{in}(1-\alpha) + L_{in} + L_{out} + H_s + H_l + R \tag{1}$$

In order to account for the impact of LAISI the algorithm implemented a radiative transfer solution for the dynamical calculation of snow albedo, $\alpha$. The algorithm builds on the SNICAR model (Flanner et al., 2007) and allows for model input variables of wet and dry deposition rates of light absorbing aerosols. Thusly, the model is able to simulate the impact of dust, black carbon, volcanic ash, or other aerosol deposition on snow albedo, snow melt, and runoff. This is the first implementation of a dynamical snow albedo calculation in a catchment scale conceptual hydrologic model and raises exciting opportunities for significant improvements in forecasting for regions that may have signficant dust burdens in the snowpack (e.g. the Southern Alps, or the western slope of the Colorado Rockies).

### 8.3    The value of snow cover products in reducing uncertainty

In operational hydrologic environments, quantification of uncertainty is becoming increasingly paramount. Traditionally, hydrologic forecasts may have provided water resource managers or production planners a single estimate of the inflow to a reservoir. This individual value often initializes a chain of models that are used to optimize use of the water resource. In some cases it may be used as input to subsequently calculate a water value for a hydropower producer giving insight into how to operate the generation resources. In other cases, the value may be provided to a flood manager, who is responsible for assessing the potential downstream flood impacts.

There is a growing awareness of the need to quantify the amount of uncertainty associated with the forecasted number. In general, in hydrologic modeling, uncertainty is driven by the following factors: data quality (both for input forcing data, as well as validation (guage) data), uncertainty associated with the model formulation, and uncertainty around the parameters selected (Renard et al., 2010). The Shyft platform aims to provide tools to facilitate rapid exploration of these factors.

In Teweldebrhan et al. (2018c) not only was parameter uncertainty explored using the well-know generalized likelihood uncertainty estimation (GLUE) methodology, but a novel modification to the GLUE methodology was implemented for operational applications. The investigation of the suitability of snow cover data to condition model parameters, required a novel approach be defined to constraining model parameters. Rather than the traditional approach to GLUE limit of acceptability (GLUE LOA), Teweldebrhan et al. (2018c) relaxed the percentage of time steps in which prediction of model realizations fall within the limits. Though this approach was found in the specific case to not lessen the uncertainty of the forecasts, it provides an forward direction whereby one can investigate more thoroughly the value of discontinuous and often sparse snow product information.

Furthering the investigation of reducing forecast uncertainty through the use of remotely sensed snow cover products, Teweldebrhan et al. (2018a) explored the implementation of Data Assimilation (DA) schemes into the Shyft framework. Given the relative availability of fractional snow covered area (fSCA), this was selected as a predictor variable. Key to this study was the development of a change point (CP) detection algorithm that allowed for the exploration of the timing aspects of uncertainty associated with the fSCA. Using fuzzy logic-based ensemble fSCA assimilation schemes enabled capturing uncertainties associated with model forcing and parameters, ultimately yielding improved estimates of snow water equivalance (SWE). The results showed that by quantifying the variable informational value of fSCA observations – as a function of location and timing windows – one can reduce the uncertainty in SWE reanalysis. In this study the LoA approach to data assimilation was introduced, and improved the performance versus a more traditional particle-batch smoother scheme. In both DA schemes, however, the correlation coefficient between site-averaged predicted and observed SWE increased by 8% and 16%, respectively for the particle batch and LOA schemes.

## 9 Discussion

### 9.1 Complexity of hydrologic algorithms

Shyft is focused on providing both hydrologic service providers and researchers a robust codebase suitable for implementation in operational environments. The design principles of Shyft are formulated in order to serve this aim. Using simple approaches in hydrological routines is a design decision related to the desire for computational efficiency. Rapid calculations are necessary in order to provide the possibility to simulate a large number of regions at high temporal and spatial resolution several times per day or on demand in order to react to updated input data from weather forecasts and observations. Hydrologic routines are therefore kept as simple as possible, but also as complex as necessary, and focus has not been on the implementation of the most advanced hydrologic routines, but on known and tested algorithms that are proven in application. Furthermore, emphasis is on portions of the hydrologic model for which data exists. For this reason, the available routines are limited in hydrologic process representation, but active community contribution is envisioned, and new functionality will be implemented when significant improvement in the scope of the targeted applications is assured. Developments aiming for increase in algorithmic complexity in Shyft undergo critical testing aiming to evaluate if the efforts goes in hand with a significant increase in forecasting per-

formance or similar advantages. Of key importance is that the architecture of the software facilitates both the testing of new algorithms and model configurations within an operational setting.

## 9.2 Multiple model configuration

A significant challenge introducing new or 'innovative' approaches exists in environments that have 24/7 up-time operations. Furthermore, there is generally a requirement to maintain an existing model configuration while exploring new possibilities. Shyft is built to facilitate the replacement of outdated operational systems in several ways. Most importantly, Shyft does not force a user to give up on certain established workflows and model configurations. Hence it is well geared toward a so-called forecast-based adaptive-management workflow to evaluate multiple pre-processing configurations of weather (see Figure 1. in Anghileri et al. (2019)).

Many classical conceptual models describe the model domain in a lumped or semi lumped fashion, such as done in the original formulation of the HBV model (Bergström, 1976) or the Sacramento Soil Moisture Accounting Model developed by the US National Weather Service. Today, both of these models are still used in private and public sectors for streamflow forecasting. The concept of *cells* in Shyft allows for equivalent model domain configurations, in which a cell may represent a basin or an elevation zone. While a user is free to configure a model in such a fashion, Shyft additionally allows for easy testing of more advanced representations of the model domain while leaving other parts of the model configuration untouched.

Another example is given by Shyft's independence from requirements towards file formats and data bases. The repository concept allows a strict separation of data sources and model, which facilitates the replacement of the forecasting model in the operational setup while leaving other parts of the forecasting system, such as databases and data storage setups, unchanged.

Moreover, the above mentioned functionalities allow, in addition to using the multiple working hypothesis through multi model support, the testing of multiple model configuration, where different combinations of input data, downscaling methods, and model algorithms can be tested.

## 10 Conclusions

This model description intends to introduce a new hydrologic model toolbox aiming for streamflow forecasts in operational environments that provides experts in the business domain and scientists at research institutes and universities with an enterprise level software. Shyft is based on advanced templated C++ concepts. The code is highly efficient and able to take advantage of modern day compiler functionality, and released Open Source in order to provide a platform for joint development. An Application Programming Interface allows for easy access to all of the components of the framework, including the individual hydrologic routines, from Python. This enables rapid exploration of different model configurations and selection of an optimal forecast model.

*Code and data availability.*  The current version of the Shyft model is available from the project website: https://gitlab.com/shyft-os/shyft under the GPLv.3 license. The documentation is available at https://gitlab.com/shyft-os/shyft-doc. The dockers for Shyft eco-system available at https://gitlab.com/shyft-os/dockers. A zenodo archive with exact version of Shyft is available with DOI: 10.5281/zenodo.3634622.

*Author contributions.*  Shyft is developed by Statkraft and the University of Oslo. The two main authors to the C++ core are Sigbjørn Helset and Ola Skavhaug with later contributions from the Open Source community, including Olga Silantyeva. Orchestration and the Python
wrappers were originally developed by John F. Burkhart with later contributions from Yisak Sultan Abdella and Felix Matt. Repositories were developed by Yisak Sultan, John Burkhart, and Felix Matt. The manuscript was written by John F. Burkhart and Felix Matt with contributions from Olga Silantyeva. The case study and examples were produced by John Burkhart and Felix Matt. All authors participated in the discussion of the paper.

*Competing interests.*  The authors declare that they have no conflict of interest.

*Acknowledgements.*  The development of Shyft is led by Statkraft AS with collaboration of the University of Oslo. Funding for primary code development is from Statkraft AS, with research activities funded through the Strategic Research Initiative LATICE at the University of Oslo and through the Norwegian Research Council Projects: 222195, 244024, and 255049.

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
