# Peer review of "Shyft v4.8: A Framework for Uncertainty Assessment and Distributed Hydrologic Modelling for Operational Hydrology"

_Geoscientific Model Development, 2020_

## Referee Comment (RC1) · Anonymous Referee #1 · 3 Jun 2020

The manuscript titled "Shyft v4.8: A Framework for Uncertainty Assessment and Distributed Hydrologic Modelling for Operational Hydrology" by Burkhart et al. describes a hydrological modelling framework for streamflow forecasting targeted for use in hydropower production and research. The authors give a detailed descriptive document of the hydrologic modelling software: Shyft, which enables the development and implementation in operational setting and capability with multiple model and forcing. In addition, they shows three applications including: i) streamflow forecasting in a Scandinavia basin, ii)the investigation of aerosol impact on the snow melt and discharge, and iii) uncertainty reducing by snow assimilation with snow cover products. The manuscript is well written, and the studies are well designed. This work is of high interest for the

hydrology community as well as for hydropower industry. I therefore recommend publication after resolving the issues / answering the questions below in the revision.

1. It is nice that authors gave a detailed statement on the reasons of building such a new hydrological framework aim for operational purpose. The authors have emphasized very much in the paper about the efficiency of the new software, but no such information was further shown in the paper. How this Shyft are superior than other existed models or framework in terms of it's computational efficiency? A bit more information and some comparison results are appreciated in the paper. 2. for the Shyft's architecture and description, the uncertainty assessment methods / components were not seen in the paper, except in one of the application papers by Teweldebrhan et al. (2018b). This needs a better clarification. 3. In the hydrology community, the regionalization is one of the main challenges regarding the prediction in Ungauged Basins (PUB). Has this been considered in the Shyft? And How the Shyft could deal with it under it's structure? 4. If I understand correctly, for the spatial interpolation, there are only IDW and Bayesian Kriging methods can be chosen in the Shyft? But how are the temperature lapse rates considered in the interpolation, which is very important for hydrological modelling, especially in glacier- / snow-fed region? 5. One of the challenges in such an operational-based hydrological forecasting framework is probably to balance between a better forecasting performance and a better computational efficiency? How is Shyft designed and deal with such conflict?

Some specific comments and technical corrections: 1. Line 69: missing a comma here. It should be : "scale, (iii)" 2. Fig. 2: the sentence in "Simulate" box can not be seen properly. 3. Acronym needs explanation for the first time. For example, in the Fig 2 and line 249: What is "PTSSK" ? 4. Line 555 and line 566: what are LOA and LoA? Are they the same thing? 5. Line487-488: "Using the model state based on the historical simulation and latest discharge observations, the model state is updated so that the discharge at forecast start equals the observed discharge. " We can see in the Fig.5 that the red line of historical simulation has shown a large bias comparing with black

line of observation (it also missed the time of peak flow). Could you explain a bit more clearly on why such a big bias and how do you exactly use the historical simulation (red) and latest discharge observation (black) for updating the model initial condition?

---

## Referee Comment (RC2) · Anonymous Referee #2 · 13 Aug 2020

[referee-annotated manuscript omitted]

---

## Author Comment (AC1) · 14 Oct 2020

**1 Response to Reviewer 1**

Dear Reviewer,

Thank you for your review.

[Figure]

1.1 It is nice that authors gave a detailed statement on the reasons of building such a new hydrological framework aim for operational purpose. The authors have emphasized very much in the paper about the efficiency of the new software, but no such information was further shown in the paper. How this Shyft are superior than other existed models or framework in terms of it's computational efficiency? A bit more information and some comparison results are appreciated in the paper

We added a section on main performance characteristics. Section 6 "Computational Performance" is added with results from a benchmarking analysis. We don't provide a direct comparison with other models, partly for the reasons described – they are not all readily available (e.g. commercial products) or it was out of the scope of the purpose of our research and development activities. However, we are unaware from experience of a hydrologic modeling system that can simulation 8000 km2 at 1km2 resolution and hourly timesteps in under 30 seconds. Please see additions in Section 6 for further details.

1.2 for the Shyft's architecture and description, the uncertainty assessment methods / components were not seen in the paper, except in one of the application papers by Teweldebrhan et al. (2018b). This needs a better clarification.

We have added Section 5.8 "Uncertainty analysis" where we discuss further the possibilities and approach one may take to evaluate uncertainty of the analysis. Details regarding the results of an analysis remain presented in Teweldebrhan et al. (2018b).

1.3   In the hydrology community, the regionalization is one of the main challenges
      regarding the prediction in Ungauged Basins (PUB). Has this been considered in
      the Shyft? And How the Shyft could deal with it under it's structure?

This is a terrific point, and one we are further exploring currently. We did bring in a
section on the topic, however, we have not thoroughly explored Prediction in Ungauged
Basins as a research exercise to date. Nonetheless, we are confident that the flexibility
afforded the modeler within Shyft presents tremendous opportunities to explore PUB
topics in detail and efficiently.

1.4   If I understand correctly, for the spatial interpolation, there are only IDW and
      Bayesian Kriging methods can be chosen in the Shyft? But how are the tem-
      perature lapse rates considered in the interpolation, which is very important for
      hydrological modelling, especially in glacier- / snow-fed region?

Thank you for highlighting this important point. We have expanded Section 5.1.3 "Gen-
eralization" and added some clarifying text. Most importantly, we point out that the IDW
routines are specified for each of the input variables. In particular, for temperature, the
modeler is offered two options:

1. The temperature lapse-rate is computed using the nearest neighbors with suffi-
   cient/maximized vertical distance.

2. The full 3d temperature flux vector is derived from the selected points, and then
   the vertical component is used.

1.5 One of the challenges in such an operational-based hydrological forecasting framework is probably to balance between a better forecasting performance and a better computational efficiency? How is Shyft designed and deal with such conflict?

This is certainly one of the primary challenges modelers face. Not only in hydrologic forecasts, but overall across a range of disciplines. Indeed, as the ability to scale simulations across cloud infrastructure and multinode architectures increases, so do costs. The aim of Shyft has been to maintain the highest possible computation efficiency *before* trying to scale out hardware – though the latter is certainly possible as well. We have added a few discussion points regarding the goals of Shyft with respect to these points in Section 5.10, "Hydrological Forecasting".

1.6 Specific comments and technical corrections

1.6.1 Line 69: missing a comma here. It should be : "scale, (iii)"

text corrected

1.6.2 Fig. 2: the sentence in "Simulate" box can not be seen properly.

believe this is fixed in our version

1.6.3 Acronym needs explanation for the first time. For example, in the Fig 2 and line 249: What is "PTSSK" ?

Figure 2 caption is updated with necessary information
**1.6.4** Line 555 and line 566: what are LOA and LoA? Are they the same thing?

The text is updated to explain LOA as limit of acceptability acronym, only LOA is used in the updated version.

**1.6.5** Line 487-488: "Using the model state based on the historical simulation and latest discharge observations, the model state is updated so that the discharge at forecast start equals the observed discharge. " We can see in the Fig.5 that the red line of historical simulation has shown a large bias comparing with black line of observation (it also missed the time of peak flow). Could you explain a bit more clearly on why such a big bias and how do you exactly use the historical simulation (red) and latest discharge observation (black) for updating the model initial condition?

We added text to clarify that "the black bar is the time-step, when the internal states updated to match observations", thus the mismatch between historical simulations and the observations is compensated.

**2  Response to Reviewer 2**

Dear Reviewer,

Thank you for the comments to our manuscript. You made several constructive and welcome comments, particularly with respect to some missing references. I acknowledge wholeheartedly our lack of reference to the HEPEX community. We have amended the manuscript to address and acknowledge more thoroughly the volume of work on the topic that has been conducted to date. Nonetheless, the intention is not to provide a review of the topic, though such a contribution would be welcome.

However, the comment: "To summarize, as a software advertisement, this can be accepted as it is, as a scientific contribution to forecasting for (hydropower) optimization this need to be rejected." seems somewhat unproductive. We do not intend that our contribution to GMD is a 'software advertisement', nor do we have the ambition to present a rigorous and novel contribution to hydropower optimization (which is actually completely off-topic for what Shyft is). Rather, our contribution is intended as a "Model Description Paper". We feel our contribution fulfills well the requirements of such a manuscript as outlined: https://www.geoscientific-model-development.net/about/manuscript_types.html#item1

**2.1  Response to comments embedded within Reviewer 2 Supplement**

**2.1.1  Title: Your manuscript is very poorly referencing the large efforts on operational hydrology stemming from the www.hepex.org community.**

Thank you for pointing this out. We are aware of HEPEX and acknowledge not adequately citing the pool of resources available within the community. We have added a paragraph specifically highlighting some of the work within HEPEX.

**2.1.2  Line 15:**

```
Pagano, T. C., and Coauthors, 2014: Challenges of Operational River Forecasting. J. Hydro
```

```
Wu, W, Emerton, R, Duan, Q, Wood, AW, Wetterhall, F, Robertson, DE. Ensemble flood foreca
```

```
Pappenberger, F., Pagano, T. C., Brown, J. D., Alfieri, L., Lavers, D. A., Berthet, L., $
```

Very relevant citations to include. Thank you.

**2.1.3 Line 25:**

```
Anghileri, D., Voisin, N., Castelletti, A., Pianosi, F., Nijssen, B., & Lettenmaier, D.

Anghileri, D., Monhart, S., Zhou, C., Bogner, K., Castelletti, A., Burlando, P., & Zappa
```

Thank you in particular for bringing our attention to the work of Anghileri et al. We have included these references.

**2.1.4 Line 37:**

```
Germann, U., Berenguer, M., Sempere-Torres, D., & Zappa, M. (2009). REAL – ensemble radar

Liechti, K., Panziera, L., Germann, U., & Zappa, M. (2013). The potential of radar-based
```

We have added these references to the QPE sentence.

**2.1.5 Line 41:**

```
Zappa, M., Rotach, M. W., Arpagaus, M., Dominger, M., Hegg, C., Montani, A., $\ldots$ Wu
```

Thank you for making us aware of the MAP D-PHASE project. Unfortunately, it appears none of the urls under map.meteoswiss.ch are valid. Nonetheless, the manuscript provides a poignant lesson regarding the challenges of implementing such a novel system in an operational setting.

**2.1.6 Line 61:**

```
Similar approach is used in the "Routing System 3.0" by EPFL, www.hydrique.ch Unfortunate
```

Yes, this is typical for much of the software we have 'heard' about, but for which we failed to find good resources, documentation, or appropriate licensing.

**2.1.7 Section 1.2 Heading:**

```
Do you know FEWS?
https://www.sciencedirect.com/science/article/pii/S1364815212002083}
```

We are well aware of FEWS. However, FEWS is not Open Source, but rather "Open Software", which presented an immediate challenge for our philosophy. More importantly, FEWS was not suitable to our requirements for further development (e.g. developing models within a common computational framework).

FEWS is really a system designed around linking different model platforms using XML configuration files. It is undoubtedly a valuable contribution to the operational community as demonstrated by the wide-spread adaptation (they even convinced the US and NOAA!). Unfortunately, we were not convinced that there would be a significant speed up in the computational component using such a system. If the software FEWS is calling is not well optimized to run in a multi-threaded or multi-node manner it will still be slow.

Lastly, to my knowledge FEWS runs on Windows. Glancing through the https://oss. deltares.nl/web/delft-fews/faqwebsite, I lack a reference to both the license they use and a simple section on 'system requirements'.

We respect the FEWS initiative and community, but did not find the product suitable for our requirements.

**2.1.8 Line 127:**

```
This would interest me
```

The intention of our manuscript is not to find failings of other platforms, softwares, or otherwise, but rather to highlight our own novel contribution to the community.

However, the key criteria we sought when evaluating other softwares included:

- Open Source License and clear License Description

- Readily accessible software (e.g. not trial or registration based)

- High quality code

    well-commented

    modern standards

    api-based, not a GUI

    highly configurable using Object Oriented standards

- Well documented software

There are likely softwares that meet these standards, particularly today, as the 'open source' movement has significantly advanced recently. However, as we started the development of Shyft, we were unable to find a suitable alternative.

We have added some further points to section 1.2.

**2.1.9   Line 140:**

```
Python is the future
```

```
https://www.youtube.com/watch?v=Og847HVwRSI
```

Thank you for the entertaining video. Though... if you were to have written this comment in 1985, "ADA would be the future" (shortlived) and in 2000 Java. The point is Python **was** the future, but now we have entered an entirely new realm of innovation with programming languages. The future, is ill-defined and I assure you, to be *code-agnostic* is the future. What about for example Julia and Rust?

Believe me, I have been a "Pythonista" for *many* years, since well before attending my first PyCon in 2012: https://us.pycon.org/2012/schedule/presentation/316/

I fully agree with the power and suitability of Python for a wide-array of activities, but I also believe in innovation.

**2.1.10  Line 152:**

```
Citations?
```

Added.

**2.1.11  Line 161:**

```
Nice
```

Thank you.

**2.1.12  Line 176:**

```
Sounds great!
```

We agree!

**2.1.13 Line 180:**

```
Reference
https://www.tandfonline.com/doi/full/10.1080/02626667.2015.1031761?scroll=top&needAcc
```

**2.1.14 Section 3 Heading:**

```
Read like a product flyer
```

Changed to "Architecture and Structure". Though, generally disagree with the comment. Understanding this is an important component of working with Shyft.

**2.1.15 Line 363:**

```
https://doi.org/10.2478/johh-2018-0004
```

Added a reference.

**2.1.16 Line 366:**

```
References!
```

Added.

**2.1.17 Line 395:**

```
This might change quite a lot from a day to the next. I am not in favor to global temperat
```

I can agree with your concern here, but this is a choice that is made only in one implementation. The point of Shyft is for users to provide their own implementations of the various algorithms that are used through the framework. We welcome your future contributions!

**2.1.18 Line 406:**

```
References!
```

fixed.

**2.1.19 Line 479:**

```
References!
```

We are not choosing to cite or specify and one particular configuration or modeling system here.

**2.1.20 Line 494:**

```
Some statistic needed!
```

We have only presented this as a 'cartoon' demonstration of the capabilities of the system. We agree a more robust description is merited, yet feel this is beyond the scope for this model description paper. We plan a forthcoming manuscript dedicated to the evaluation of the ensemble streamflow prediction capabilities of Shyft.

**2.1.21 Line 497:**

How?

Agreed that this could be quantified better. However, unfortunately at present, the authorship team has limited ability to evaluate the forecasts in further detail.

**2.1.22 Line 500:**

I know, there is also large research on it!

Anghileri, D., Monhart, S., Zhou, C., Bogner, K., Castelletti, A., Burlando, P., & Zappa,

Included earlier.

**2.1.23 Section Heading 7.3:**

Is this now implemented in shift or is this a general information? We need examples.

Yes, please see the Teweldebrhan references.

**2.1.24 Line 548:**

Yes, and there is huge literature on this.

Yes, and a topic in it's own right!

**2.1.25 Line 603:**

Product factsheet

We have changed to "model description" paper.